# Obesogens in Foods

**DOI:** 10.3390/biom12050680

**Published:** 2022-05-09

**Authors:** Iva Kladnicka, Monika Bludovska, Iveta Plavinova, Ludek Muller, Dana Mullerova

**Affiliations:** 1Department of Public Health and Preventive Medicine, Faculty of Medicine in Pilsen, Charles University, 301 00 Pilsen, Czech Republic; monika.bludovska@lfp.cuni.cz (M.B.); iveta.plavinova@lfp.cuni.cz (I.P.); dana.mullerova@lfp.cuni.cz (D.M.); 2Department of Cybernetics, European Centre of Excellence New Technologies for the Information Society, University of West Bohemia, 301 00 Pilsen, Czech Republic; muller@kky.zcu.cz; 3Institute of Pharmacology and Toxicology, Faculty of Medicine in Pilsen, Charles University, 301 00 Pilsen, Czech Republic

**Keywords:** obesity, obesogens, food, adipose tissue, metabolic disruptors, systematic low-grade inflammation, metabolic syndrome

## Abstract

Obesogens, as environmental endocrine-disrupting chemicals, are supposed to have had an impact on the prevalence of rising obesity around the world over the last forty years. These chemicals are probably able to contribute not only to the development of obesity and metabolic disturbances in individuals, but also in their progeny, having the capability to epigenetically reprogram genetically inherited set-up points for body weight and body composition control during critical periods of development, such as fetal, early life, and puberty. In individuals, they may act on myriads of neuro-endocrine–immune metabolic regulatory pathways, leading to pathophysiological consequences in adipogenesis, lipogenesis, lipolysis, immunity, the influencing of central appetite and energy expenditure regulations, changes in gut microbiota–intestine functioning, and many other processes. Evidence-based medical data have recently brought much more convincing data about associations of particular chemicals and the probability of the raised risk of developing obesity. Foods are the main source of obesogens. Some obesogens occur naturally in food, but most are environmental chemicals, entering food as a foreign substance, whether in the form of contaminants or additives, and they are used in a large amount in highly processed food. This review article contributes to a better overview of obesogens, their occurrence in foods, and their impact on the human organism.

## 1. Introduction

The rapid and significant increase in the prevalence of obesity worldwide over the last forty years is considered not to be attributed solely to genetic or life style risk factors, such as energy-dense and nutritionally poor diets, sedentary lifestyle, or aging. New evidence has shown that epigenetic, central regulatory pathways, and endocrine-disrupting changes that are associated with human exposure to man-made chemicals might also contribute to the obesity epidemic. So-called obesogens are xenobiotics directly or indirectly promoting adipogenesis and obesity in animals and humans, influencing individuals or their progeny. Many of these chemicals may also crossroad or modulate the effect of endogenous ligands of nuclear or non-nuclear transcription factors, participating in differentiation, metabolism, and the secretory function of adipocytes [1].

There are a number of examples in medicine that synthetically produced chemicals (drugs) may influence the development of adiposity. This usually adverse effect of pharmaceuticals is evidenced in glucocorticoids, estrogens, some antidiabetics (such as insulin, sulphonylureas, thiazolidinediones, glitazones), thyreostatics, dopaminergic blockers, beta sympathetic blockers, and, in some drugs, from the groups of tricyclic antidepressants, selective serotonin re-uptake inhibitors, atypical antipsychotic medicines, antiepileptics, neuropeptides, and eutonics of the gastrointestinal tract [2,3,4,5,6,7,8]. However, not only medicines, but many compounds introduced in mega doses to the environment over the last decades by human production, were recognized to be able to act as obesogens. The main route of human exposure is dietary ingestion through contaminated food [9,10,11]. Also in the last 40 years, the dramatically developing food industry, using new technologies in the production of highly processed foods, can contribute to the development of obesity by changing the quality of food and the increased content of certain nutrients or additives [9].

## 2. Materials

We followed the current methodological guidelines for systematic reviews to identify, retrieve, and summarize the relevant epidemiological literature on the relation between obesogens and overweightness/obesity, Type 2 diabetes, metabolic syndrome, and atherosclerotic cardiovascular disease [12,13]. Each eligible paper was summarized with respect to the methods and results, with particular attention paid to the study design and exposure assessment. All articles were searched using Medline and Web of Science; we focused on the original articles and excluded doubled articles. We used the following search terms: “obesogens, metabolic disruptors, obesogens in food, food, additives, contaminants, obesity, adipose cells, adipose tissue, metabolic syndrome, systematic low-grade inflammation”.

## 3. Obesity

Pre-obesity (overweight) and obesity are medical conditions marked by an abnormal and/or excessive accumulation of body fat that presents a risk to health (WHO 2019). According to the last definition, adopted by the European Commission in 2021, obesity is a chronic, relapsing disease, which in turn acts as a gateway to a range of other non-communicable diseases, such as diabetes, cardiovascular diseases, and cancer.

The obesity prevalence has risen exponentially in the world’s population over the last 40 years. While in 1975, 6.4% of women and 3.2% of men were obese, the prevalence by 2014 roughly tripled to 14.9 and 10.8%, respectively. According to a prediction, every fifth adult will suffer from obesity in 2025. The global age-standardized mean body mass index (BMI) of children and adolescents aged 5–19 years has also been increased during the evaluated period from 1976 to 2016 in both genders, leading to virtually identical age-standardised mean BMIs for both genders [14].

The obesity pandemic has been probably brought about by dramatic changes in lifestyle during a relatively short period of human evolution. This maladaptation is the result of complex interactions between biological, behavioral, social, and environmental factors that are involved in the regulation of energy balance and fat stores.

In addition to increased mechanical load on the musculoskeletal system and cardiorespiratory load, obesity is a metabolic disease that is associated with dysfunctional white adipose tissue, affected by systematic low-grade inflammation. This leads to chronic systemic inflammation, ectopic fat accumulation in tissues and organs, a pro-coagulative state, endothelial dysfunction, and impaired carbohydrate, lipid, protein, and purine metabolism. It is linked to clinical conditions, such as hypertension, dyslipidemia, Type 2 diabetes mellitus, cardiovascular, and tumor diseases.

The reported estimates for the population-attributable risks of obesity have been shown to range from 5 to 15% for all-cause mortality, from 0.2 to 8% for all-cancer incidence, from 7 to 44% for cardiovascular disease incidence, and from 3 to 83% for Type 2 diabetes mellitus incidence [15]. Obesity is one of the leading causes of death and disability worldwide and is the fourth highest independent cause or premature mortality.

The histopathological unit of dysfunctional adipose tissue is characterized by adipocyte hypertrophy with infiltration of M1 macrophages, as well as impaired adipogenesis, angiogenesis, lipolysis, and de novo lipogenesis in adipose tissue.

## 4. Adipose Tissue

Adipose tissue is a complex, heterogeneous, and highly dynamic organ, executing the storage of energy and contributing to the control of energy metabolism of the whole organism. It consists of specific cells—maturate adipocytes that are differentiated under endocrine stimuli from their mesenchymal stem cell precursors during adipogenesis.

According to the morphology and function of the predominant maturated adipocytes three types of adipose depots are recognized in humans: the white (WAT), the brown (BAT), and the beige/brite/brown-like (BAT) adipose tissues. WAT contains adipocytes with a single large unilocular lipid droplet filling most of the cytoplasm and pushing the nucleus and organelles to the margins of the cells. BAT is characterized by smaller-sized adipocytes, with an abundance of smaller lipid droplets and many mitochondria. The third type—BAT—represents a combination of the attributes of previous two. Beige adipocytes are of middle size, have more lipid droplets and fewer mitochondria than BAT. It is supposed that WAT can be transformed into beige adipocytes under thermogenic stimuli [16].

Mature adipocytes are able under neuroendocrine control to store energy in lipid droplets in the form of triglycerides, and release it in the chemical (WAT) or thermal form (BAT) according to the body’s requirements. The remaining cells are made up of stromal vascular fraction and belong to the immune, epithelial, vascular, and stromal cells. Besides storage and distribution of energy, adipose tissue contributes to the regulation of systemic energy metabolism by the secretion of adipokines that enables endocrine, paracrine, autocrine, and cross-talk communication with other organs. The physiological production of adipokines requires intact cellular machinery of mature adipocytes, in particular mitochondrial respiration and balance between lipogenesis and lipolysis. As hormones regulate the physiology of these systems, their action can be disrupted by chemicals in the environment that mimic or block normal endocrine functions [17].

Dysregulation of adipocytokines caused by obesity contributes to the pathogenesis of various metabolic and cardiovascular disorders [18].

## 5. Etiology of Obesity

Obesity and related disorders have become a public health issue [19,20]. As a multifactorial disorder, obesity cannot be linked specifically to one etiology, including genetics or environmental chemicals. While dietary restriction and increased exercise continue to be the most prescribed treatment, the obesity pandemic continues unabated and is increasing worldwide [21]. Despite the voluminous literature on obesogens and metabolism-disrupting chemicals, a series of workshops aimed at identifying the best evidence for the effects of these factors on obesity and diabetes have identified shortcomings in the available data that have prevented a complete and accurate analysis of their impact.

Obesity is most likely caused by (1) imbalance between energy intake and expenditure, resulting in energy surplus (e.g., by consumption of high-calorie diets); (2) genetic predisposition (40–70%), as well as hormonal, environmental, biological, psychological, and sociological factors; (3) lack of physical activity; (4) exposure to obesogens (endocrine disruptors or diabetogens) [22].

## 6. Obesogens

### 6.1. The Obesogen Hypothesises

The possible impact of obesogens, originally a highly controversial issue, has been supported by a growing body of evidence. Obesogens include xenobiotics that promote adipogenesis and obesity in animals and humans, such as several medicines or substances acting as endocrine disruptors [23]. Human activities have polluted water, soil, and foods. Obesogens are currently contained in many products for daily use, e.g., personal care products, cosmetics, cleaners, toys, kitchen utensils, plastic curtains and table cloths, soft furnishings, furniture, mattresses, and clothes.

Obesogens are chemicals that directly or indirectly increase fat accumulation and cause obesity [24]. The obesogenic hypothesis further suggests that obesogens can act directly at the cellular level to increase the commitment or differentiation of adipocytes from stem cells by altering the number of adipocytes, increasing the retention of triglycerides within adipocytes, or modifying the rate of adipocyte proliferation when compared to cell death. Furthermore, obesogens can act indirectly as well by changing basal metabolic rate, shifting energy balance to favor calorie storage, and modulating food intake and metabolism via effects on the adipose tissue, brain, liver, pancreas, muscle, and the gastrointestinal tract [21,22].

To sum up, obesogens promote adipogenesis and fat accumulation, affect appetite control and satiety, and act as endocrine disruptors, possibly changing hormonal regulations [25].

The effects of obesogens can only become apparent later in life [26]. Previous studies have identified obesogens that have the potential to disrupt multiple metabolic signaling pathways in the developing organism, resulting in permanent changes to the adult’s physiology. Prenatal or perinatal exposure to obesogenic endocrine-disrupting chemicals has been shown to predispose an organism to store more fat from early life [27].

This suggests that humans, who have been exposed to obesogenic chemicals during sensitive periods of development, might be pre-programmed to store increased amounts of fat, resulting in a lifelong struggle to maintain a healthy weight [24].

In this case, obesogens alone do not cause obesity in humans, but can work behind-the-scenes to promote weight gain, due to the developmental programming of adipose tissue regulation, poor diet, and metabolism [28].

In 2019, a study by Heindel and Blumberg provided strong evidence of the presence of estrogens acting as obesogens in humans. Since 2009 (the study of Newbold et al.), it has been known that the same holds for animals. In previous years, studies have identified transcriptomic and metabolomic changes of polychlorinated biphenyl-126 (PCB-126) in human hepatocytes, HepaRG, that imply the possibly detrimental role of environmental pollutants for the development of non-alcoholic fatty liver diease (NAFLD). These impacts might be precipitated by poor diet and/or a sedentary lifestyle [29]. Biological mechanisms acting in the development of hepatic steatosis are divided into four categories: increased fatty acid uptake, decreased lipid efflux, increased fatty-acid synthesis, and impairment of the oxidative metabolism of these substances [30]. The further elucidation of impaired hepatic lipid metabolism is needed [31]. Animal studies have clarified the impact of obesogens on the etiology of obesity. Currently, they focus on human lipid metabolism. Tissue culture studies are being carried out predominantly on 3T3-L1, derived from mouse cells. The main aims of 3T3-L1 studies are the clarification of the obesogenic potential of xenobiotics and their metabolites, as well as the assessment of their impact on adipose differentiation. Xenobiotics may increase the number of differentiated 3T3-L1 pre-adipocytes and enhance their capacity for droplet storage. The mechanism behind is most probably the up-regulation of transcription factors CCAAT/enhancer-binding protein α (C/EBP α) and peroxisome proliferator-activated receptor ɣ (PPARɣ). These are associated with significantly higher expression of fatty-acid-binding protein 4 adipokine [32].

PPAR is a ligand-activated transcription factor, which is responsible for the growth and development of adipose tissue and that acts as the receptor for antidiabetic drugs such as rosiglitazone [33,34]. Neither the mechanism nor the modification of the key cellular processes lying between induction of the receptor and onset of the disease have been described [35,36].

The most common contaminants that are considered to be potential obesogens include estrogens, such as diethylstilboestrol and genistein; organotins, e.g., tributyltin; fluoro actanoates; bisphenol A; diethylhexyl phthalate. These chemicals directly alter endocrine function and metabolic organs that control lipid homeostasis (e.g., the liver), suggesting that exposure might be a risk factor for the development of NAFLD [22,23,37,38,39].

In 2015, the Parma consensus broadened the definition of obesogens to include endocrine disrupting chemicals that affect other obesity-related metabolic conditions that drive metabolic syndrome, such as insulin resistance, hypertension, dyslipidemia, and hyperglycemia [17]. This class of endocrine-disrupting chemicals was denoted as being metabolism-disrupting chemicals [40].

### 6.2. Overview of Obesogens

Presence in foods

***1.*** 
**
*Naturally occurring obesogens*
**
FructoseGenistein
***2.*** 
**
*Xenobiotics*
**
**2.1.** 
**Contaminants**
PharmaceuticalsDiethylstilbesterolEstradiolRosiglitazoneOrganic Pollutants (OP’s)**Industrial Chemicals**Bisphenol A (BPA)OrganotinsPerfluorooctanoic Acid (PFOA)PhthalatesPolybrominated Diphenyl Ethers (PBDEs)Polychlorinated Biphenyl Ethers (PCBs)**Organophosphate Pesticides**ChlorpyrifosDiazinon**Organophosphate Pesticides**Dichlordifenyltrichloretan (DDT),Dichlordifenyltrichloretan (DDT),**Other Environmental Pollutants**Benzo[a]pyreneFine Particulate Matter (PM_2.5_)Triclosan
**2.2.** 
**Additives**



**1.** 
**
*Naturally occurring obesogens*
**


Fructose

Fructose, a monosaccharide present in fruits and honey, promotes the development of obesity easier than glucose. Its overconsumption contributes to the increasing prevalence of obesity, insulin resistance, and metabolic as well as cardiovascular diseases [41]. Fructose is capable to affect the intestinal microflora with increased intestinal permeability [42]. Fructose-2,6-bisphosphate derived from fructose-6-phosphate has been identified as one of the signaling metabolites responsible for glucose-induced recruitment of carbohydrate response element binding protein (ChREBP) to its target genes. ChREBP promotes de novo lipogenesis in liver and adipose tissue [43,44].

Due to the different metabolism and high lipogenic potential by fructose when compared to glucose, fructose ingestion precipitates the accumulation of excessive fat in the liver and results in weight gain and abdominal obesity [45].

Recently, fructose has become overabundant in the food industry, especially in the case of non-alcoholic sweetened beverages and sweets.

Genistein (in soy)

Phytoestrogens, contained in various foods and food supplements, in particular soy products, are another prominent class of chemicals. Genistein and daidzein are two of the most abundant phytoestrogens in the human diet. For its estrogenic activity, genistein has been proposed to have a role in preserving good health by regulating lipid and carbohydrate homeostasis [46]. Genistein is also used as a supplement for menopausal woman. However, a recent study showed that only at high doses did genistein indeed inhibit adipose deposition, but, at low doses similar to that found in Western and Eastern diets, in soy milk or in food supplements containing soy, it surprisingly induced adipose tissue deposition, especially in males. Further, this increase in adipose tissue deposition by genistein was correlated with mild peripheral insulin resistance. Interestingly, genistein did not significantly affect food consumption [47] suggesting an abnormal programming of factors involved in weight homeostasis [48].

**2.** 
**
*Xenobiotics*
**


Xenobiotics are not natural compounds found in foods and are not used as separate foods. The presence of xenobiotics in foods, according to the dose, can be harmful to humans.

Substances that are not naturally occurring compounds of foods are called foreign substances. Foreign substances or xenobiotics are classified either as contaminants or as additives.

   **2.1.**  **Contaminants**

Substances contaminating foods unintentionally, not posing a risk in usual concentrations but being potentially harmful at higher doses, are called contaminants. They can contaminate food. Contamination may occur at each step of the production chain.

The most common causes of contamination are: the use of veterinary drugs, contaminated soil from environmental pollution, persistent organic pollutants for agricultural purposes, sanitary materials, radioactive contaminants, traffic pollutants, and contamination from packaging materials.

Substances contained in packaging materials, such as phthalic acid, are used as a softener.

The primary contaminants of high concentration include toxic metals, above all being lead, mercury, cadmium, and inorganic chemicals, e.g., nitrous and nitric oxide.

       2.1.1.  Pharmaceuticals

Some pharmaceuticals used in veterinary medicine and in animal production may act as possible obesogens in humans:

Diethylstilbesterol (DES)

This is an estrogen that was prescribed to millions of women from 1940–1971 to prevent abortion in the first trimester of pregnancy. The prescription has been suspended due to adverse side effects, but the drug is still being used to enhance fertility in livestock and, therefore, enters the food chain. DES may have acted an obesogen in the human population [49].

       2.1.2.  Organic pollutants (OPs)

These toxic and cancerogenic chemicals are very resistant to degradation and many of the products of their decomposition are toxic as well. The greatest risk stems from their ability to accumulate in the food chain. The main source of OPs are animal foods (meat, fatty fish, dairy products, and eggs).

           2.1.2.1.  Industrial chemicals

Bisphenol A (BPA)

BPA is one of the highest-volume chemicals used in commerce. Its omnipresence in polycarbonate plastics, epoxy resins (automobile parts, safety protective equipment, food and water containers, baby bottles, or the protective lining inside metal food cans, dental fillings, etc.), and thermopaper contributes to continuous human exposure [50,51]. Dietary ingestion is suspected to be the main route for human exposure, although dermal exposure can also occur from skin contact with thermal paper. BPA has been detected at measurable concentrations in the urine samples of almost all persons tested worldwide. In addition, BPA has been detected in placental and amniotic fluids and human breast milk (Blumberg 2021).

BPA is an endocrine disruptor exhibiting estrogen-like activity that is able to affect the regulation of leptin and insulin production, and thus acts as an agonist and antagonist of PPARy [52].

Many studies clearly support the enhancement of adipogenesis, dysregulation of adipocytes and glucose, and the inflammatory changes of adipose tissue resulting from BPA, resulting in obesity [53,54].

A systematic review with a meta-analysis of the epidemiological evidence, given by Wu et al., has revealed a positive correlation between the level of BPA and obesity risk. A dose–response analysis revealed that a 1 ng/mL increase in BPA increased the risk of obesity by 11%. There were similar results for different types of obesity, gender, and age [55].

Due to its adverse effects on human health, the European food safety authority (EFSA) has determined the tolerable daily intake of BPA (4 μg per kilogram of body weight per day) [56]. Today, there is a growing tendency to replace BPA with its analogues. This is based on legal limits for BPA in basic goods. In 2019, a longitudinal cohort study revealed a significant association of bisphenol S (BPS) and bisphenol F (BPF) with obesity in children aged 6 to 19 when compared to total bisphenol and BPA. The replacement of BPA with other bisphenols therefore might not be efficient [57].

Organotins (OTs)

These are chemicals widely used as pesticides, disinfectants, and biocides in paints. OTs are harmful to endocrine glands and can interfere with neuroendocrine control, hormone synthesis and/or the biological availability or activity of target receptors. They impair metabolism either centrally (lateral hypothalamus) or peripherally (adipose tissue) and result in obesity. Besides their obesogenic effects, OTs affect reproductive organs [58,59]. Due to their physical and chemical properties, OTs easily enter food chains and produce tributyltin (TBT) and triphenyltin, which are both severely toxic [60,61].

The most common OTs include TBT. Pilot studies have shown a positive correlation between placental TBT concentration and weight gain in infants [62,63]. Humans are exposed to TBT in seafood, foods treated with agricultural fungicides and miticides, industrial waters, textile material, polyvinyl chloride stabilized with TBT, or food packaging. Indirectly, house dust, which contains significant amounts of TBT, can be the source of contamination [64,65].

Perfluorooctanoic Acid (PFOA)

This is a group of synthetic chemicals that is used for their high resistance and stability. They are intermediates of Teflon production, and are commonly found in the environment. The route of exposure is mostly the digestive tract.[4] Evidence supports the obesogenic effect of PFOA, though its biological mechanism needs further clarification. A study by Lia et al. (2020) demonstrated that the obesogenic effect of PFOA was the result of a combination of many enzymatic pathways with insulin signaling [66]. In 2022, a study supporting previous findings suggested that PFOA might act as a developmental obesogen, transmitted vertically via the placenta [67].

Phthalates

Phthalates or phthalic esters form a group of chemicals that are used as softeners for plastics, additives for cosmetics, insecticides, or as adhesives. They can be detected in breast milk and enter foods from packaging materials, including package water and spirits, but they are particularly present in fatty foods, because they are lipophilic. In addition, toys containing phthalates can enter the organism when placed into the mouth [68]. Phthalates are one of the most studied metabolic disruptors. Several observational studies suggested that phthalates could be determinant in the pathogenesis of obesity [69]. Phthalates act as thyroid hormone agonists as well as androgen agonists. Thus, they can affect adipogenesis, fat accumulation, and insulin resistance by interfering with PPAR activation [70]. A recent study from 2020 suggests an association of child growth with prenatal exposure to phthalates, especially those of low molecular weight.[71] Furthermore, in the case of chronic exposure to low doses of phthalates, adverse effects (spermiotoxic, embryotoxic, and teratogenic) on the reproductive system were observed, as well as hepatotoxicity and nephrotoxicity [68,72].

Polybrominated Diphenyl Ethers (PBDEs)

PBDEs were used as flame retardants in plastics, electronics, vehicles, households, furniture, textile material, and building materials. Several studies showed an association between PBDEs and foods, as they were detected in butter, fish, and other products high in animal fat [73]. Despite a production ban due to adverse effects on human health, the use of reserves is still allowed. One of the most common PBDEs is PBDE 99, which can be detected the most in adipose tissue, especially white adipose tissue [74,75]. An epidemiological study demonstrated a positive correlation between early exposition to PBDEs and increased adiposity at the age of 8 years [76]. This has been confirmed at the cellular level with PBDE 99 and the adipocyte lineage of C3H10T1/2 [77]. These studies support the obesogenic effect of PBDEs. In addition to a pro-adipogenic effect in cell cultures 3T3-L1, they increase fat accumulation, as well as C/EBAα and PPARγ expression, in the course of differentiation [78].

Polychlorinated Biphenyl Ethers (PCBs)

This is a group of fat-soluble chemicals that differ from each other in the number and position of chlorine bound to the biphenyl. There are 2010 PCB congeners. Due to their industrial use in paints, varnishes, plastics, pesticides, and coolants, they have entered the environment and foods. Long-term consumption of food containing high amounts of PCBs might be hazardous [26,79]. These are especially milk, fish, and animal foods. Animal products are contaminated via agricultural premises that have not removed paint used before 1986, before the use of PCBs was banned [80]. Animal studies suggest that PCBs promote the differentiation of adipocytes and PPAR expression, resulting in weight gains in offspring [25]. The obesogenic effect of PCBs is discussed in many studies [81]. Valvi et al. demonstrated that PCB concentration in cord blood was associated with BMI and overweightness in children at the age of 5, 6.5, and 7 years, showing a more profound effect in girls [82,83].

           2.1.2.2.  Organophosphate Pesticides (OPPs)

More than 100 various organophosphates have been described. OPPs pertain to the most commonly used pesticides worldwide and their use in agricultural premises has rapidly increased. The World Health Organization (WHO) has designated OPPs as being extremely hazardous [84,85].

Chlorpyrifos (CPF)

This is an organophosphate pesticide widely used in agriculture and, therefore, has entered the environment. Today, studies on mice 3T3-L1 models are being carried out and a study by Blanco et al. suggests that CPF and its metabolite 3,5,6-trichlorpyridinol (TCP) affect metabolism during adipogenesis, by increasing the number of differentiated 3T3-L1 adipocytes and the capacity for storage of lipid droplets. This process is linked to an up-regulation of the transcription factors CCAAT/enhancer binding protein α (C/EBP α) and PPARγ, which is accompanied by a significantly higher expression of fatty acid-binding protein 4 (FABP4) adipokine [86,87].

Diazinon

This pesticide and nematocide was widely used in agriculture and commonly detected in the human population. Residues of diazinon were also detected in ground water and drinking spring water [88]. Via inhibition of acetylcholinesterase, diazinon elicits neurotoxicity. Its pro-adipogenic effect has been shown in a study on mice 3T3-L1, where the accumulation of lipid droplets and the activation of proadipogenic signaling pathways were related to the concentration of diazinon. Diazinon significantly induced the protein expression of the transcription factors CCAAT-enhancer-binding proteins α (C/EBP α) and PPARγ, as well as their downstream proteins, fatty-acid synthase (FASN), acetyl CoA carboxylase, lipoprotein lipase, adiponectin, perilipin, and fatty-acid binding protein 4 (FABP4) [89].

           2.1.2.3.  Organochlorinated Pesticides (OCPs)

Dichlordifenyltrichloretan (DDT),

Dichlorenthylendichlordiphenyldichlorethylen (DDE)

The insecticide DDT was used on a large scale from 1939 against mosquitos *Anopheles funestus,* a vector of malaria. After its toxicity had been demonstrated, DDT was banned for use. There are exceptions to this rule, especially in developing countries that are fighting malaria. Problems with DDT and its products of degradation have continued until today because of its continuous presence in the environment. Due to its persistence, DDT has entered the food chain and has often been detected in animal adipose tissue and water. The largest part of DDT and its metabolites enter the human organism via the consumption of meat, dairy products, and fish. Leafy vegetables are usually richer in DDT when compared to other kinds of vegetables. Breast-feeding is another important form of human exposure. A growing amount of epidemiological evidence, in both in vivo and in vitro studies, have associated persistent organic pollutants, such as DDT and the DDT metabolite p,p’-DDE, with obesity [90,91,92,93]. Acute exposure causes harm to the central nervous system, while chronic exposure can result in liver cancer, disruption of endocrine control, harm of the fetus and fertility, and increased risk of Type 2 diabetes. DDT and especially its metabolite DDE may pose a risk of developing obesity in later life [94]. Rodents exposed to DDT during prenatal life have been found to have decreased energy expenditure along with glucose intolerance, dyslipidemia, and hyperinsulinemia [95].

           2.1.2.4.  Other Environmental Pollutants

Benzo(a)pyrene

This is a polycyclic aromatic carbohydrate present in smoke and a proven carcinogenic chemical that is produced in the course of burning, grilling, or smoking foods [96]. Its anti-adipogenic effect via the aryl hydrocarbon receptor has been demonstrated on cell cultures of human preadipocytes [97,98].

Triclosan

This is a commonly used antibacterial agent, present in oral care waters, toothpastes, toothbrushes, antibacterial soap, washing powder, and kitchen breadboards. Exposure to triclosan and triclocarban has been linked to an elevated risk of child obesity [99]. Animal studies show a correlation between high levels of triclosan and estrogens, androgens, and thyroid hormones. Human stem cell culture models have demonstrated an anti-adipogenic effect, including a lower production of adiponectin and lipoprotein lipase (i.e., markers of cellular fat), which is correlated to the concentration of the chemical [100].

Fine Particulate Matter (PM_2.5_)

In previous decades, particulate pollution has become a growing health issue worldwide, especially in the northern and north-western regions of China [101]. Motor transportation has considerably contributed to the concentration of PM in urban areas, as well as biomass, other waste, or industrial burning or road dust [102]. Several studies have shown an association between air pollution and the risk of obesity, predominantly in the male population [103].


**   2.2.  Additives**


Commonly used additives have been linked to obesity. These substances include the emulsifiers carboxymethylcellulose and P-80, the surfactants DOSS and Span-80, the preserving agent 3-tert-butyl 4-hydroxyanisol (3-BHA), artificial sweeteners, and the flavor enhancer monosodium glutamate (MSG) [104,105,106]. More than 350 additives have been approved in the European Union and we have focused only on examples of the most discussed.

Monosodium glutamate

This is a chemical eliciting the secretion of glucagon-like peptide-1, a hormone controlling appetite and satiety, and/or antagonization of the androgen receptor [107,108]. That MSG contributes to the early onset of obesity has been demonstrated in animal studies. Mice administered with MSG postnatally showed a significantly increased proportion of fat in both sexes. MSG administered postnatally to mice acted as a neurotoxic agent on the hypothalamic arcuate nucleus, leading to obesity [109,110].

These findings represent a promising outlook for future research, as they draw attention to the consequences of a highly processed diet.

Carrageenan

This is a hydrocolloid substance, commonly present in chocolate milk and ice cream, that is able to impair glucose tolerance, increase insulin resistance, and inhibit insulin signaling in in vivo mouse liver cells and human HepG2 cells. A study on mice from 2021 showed a significant change in gene expression related to lipid metabolism, especially in the decreased gene levels of adipocytokines, lipogenesis, lipid absorption, and transport, and the increased genes for adipolysis and oxidation after carrageenan exposure [111,112].

Antioxidants

Foods often contain antioxidants, such as the preserving agent natrium sulphite, natrium benzoate, natural coloring agents, and curcumin. Oxidative stress, caused by the consumption of additive artificial antioxidants in foods at a younger age, has been associated with the development of adiposity in later life.

Lower leptin secretion in mouse adipocytes 3T3-L1 after incubation with LPS mimicked inflammation in obesity, i.e., consuming antioxidant additives might lead to lower leptin secretion and contribute to the obesogenic environment [113,114].

All mentioned obesogens have been shown in Table 1.

## 7. Obesogen Elimination Method

The common occurrence of obesogens with which humans are in regular contact should be limited. Because humans are already heavily exposed to environmental obesogens in the form of plastics, pesticides, herbicides, industrial products, and personal care products, compounds intentionally added to foods, such as certain artificial sweeteners, phytoestrogens, preservatives, added sugars (e.g., corn syrup with a high fructose content) deserve special attention. Furthermore, it has already been proven that many obesogens are found in animals and their products, which we then use as food [106,107,108,109,110,111].

Another way to eliminate obesogens is to consume organic products that are not treated with pesticides, fungicides, and other sources of obesogens. Fruits and vegetables are commonly treated with fungicides that have been identified as obesogens, such as glyphosate used on corn, wheat, and rice [122,123,124].

Food processors also deal with the issue of obesogens. Studies are being sought looking for methods of eliminating obesogens from food. For example, Rezaei et al. published a study in 2021 to remove the pesticide diazinon from apple juice and found a successful solution using fermentation with the cultivated bacterium *Lactobacillus acidophilus*, where the product is stored in the cold for 28 days after fermentation and the diazinon is completely removed [125].

## 8. Conclusions

The study of obesogenic compounds in food is still in its early phase, and people are constantly exposed to obesogens, either directly from food or contaminated food. Regarding the objectives of food industry technologies, i.e., the extension of expiration dates, cost reduction, the best attainable palatability, optimization of production effectiveness, and food safety in terms of the absence of pathogens, over 4000 new substances have entered into foods. In most of the new substances, their impact on overall metabolic homeostasis remain unknown. Many of these xenobiotics may, as obesogens, cause epigenetic, central regulatory pathway, and endocrine-disrupting changes, directly or indirectly promoting adipogenesis and obesity in humans, influencing individuals or their progeny. Many of them may also crossroad or modulate the effect of endogenous ligands of nuclear or non-nuclear transcription factors, participating in differentiation, metabolism, and the secretory function of adipocytes. Metabolism-disrupting chemicals may affect other obesity-related metabolic conditions that drive metabolic syndrome, such as insulin resistance, hypertension, dyslipidemia, and hyperglycemia.

Confirmation of the effect of obesogens at the current exposure concentrations for the general population still requires a larger number of scientific studies to support better management of these chemicals in our environment, and to decrease human exposure.

The biological effects of main obesogenic candidates can be correctly analyzed to obtain data to advocate for the requirement to revise their regulation. Efforts should be made to better regulate the production of these obesogens and metabolic disruptors, their use and, therefore, decrease environmental and food contamination. In addition, new approaches and ways to minimize their obesogenic, and especially metabolic-disrupting, potential in humans should be under investigation, which could help to develop an efficient strategy to reverse the increasing trend of the obesity pandemic.

## Figures and Tables

**Table 1 biomolecules-12-00680-t001:** Obesogens and their effect on human health.

			Obesogens	Obesogenic Effect	Health Impact
Naturally occurring obesogens			Fructose	Different fructose metabolism and high lipogenic potential = excessive fat storage in the liver, weight gain of visceral adipose tissue. In fetal, neonatal and infant development, high exposure to fructose as an obesogen can affect lifelong neuroendocrine function, appetite control, eating behavior, adipogenesis, fat distribution [45].	obesity, insulin resistance, metabolic and cardiovascular diseases
			Genistein	At high (pharmacological) doses it inhibits adipose tissue deposition, but at low doses (normal concentration in soy) it induces adipose tissue deposition, especially in men. The genistein regulate estrogen and progesterone receptors [47].	Obesity, mild peripheral insulin resistance
Xenobiotics	Contaminants	Pharmaceuticals	Diethylstilbesterol	Endocrine disruptor with abnormal programming of various differentiating estrogen-target tissues [49].	Potential obesogen
			Estradiol	Estradiol in combination with a diet rich in fats and sugars causes variability in estrogen-induced gene expression in the dorsal raphe [7].	Potential obesogen
			Rosiglitazone	Rosiglitazone reduces hyperlipidemia and hyperglycemia, improves insulin sensitivity and decreases serum lipids, but does increase adipogenesis and lipid accumulation in tissues including liver triglyceride accumulation and hepatic steatosis [6].	Potential obesogen
		Industrial chemicals	Bisphenol A (BPA)	Endocrine disruptor, it is able to affect regulation of leptin and insulin secretion (PPARy agonist and antagonist) [50].	Supports adipogenesis, dysregulation of adipocytes and glucose, inflammation of adipose tissue → obesity
			Organotins (OTs)	OTs can damage the endocrine glands, interfering with neurohumoral control of endocrine function involves changes in the mechanism of adipose tissue [58].	Predisposition to obesity, metabolic disorders, and effects on reproductive organs
			Perfluorooctanoic Acid (PFOA)	PFOA can cause aberrant lipid metabolism in male offspring, insulin resistance, non-alcoholic fatty liver disease, with influencing PPARy signaling pathway [66,115].	Obesity, hepatic inflammation, disorders of lipid metabolism, disruption of gut barrier integrity in male offspring
			Phthalates (di-(2-ethylhexyl) phthalate (***DEHP***), di-butyl phthalate (***DBP***), benzylbutyl phthalate (***BBP***) and possibly also di-isononyl phthalate (***DINP***), di-isodecyl phthalate (***DIDP***) and di(n-octyl)- phthalate (***DNOP***)	Phthalates can cause insulin resistance, increase endoplasmic reticulum expression, disruption of glucocorticoid signaling in mesangial cells and preadipocytes [70].	Predisposition to obesity and metabolic diseases can influence metabolic regulation by disrupting the homeostasis of thyroid hormones
			Polybrominated Diphenyl Ethers (PBDEs)	PBDEs are insulin disruptors, isoproterenol stimulates the metabolism of adipocytes [116,117,118].	Predisposition to obesity, insulin resistance in obese individuals
			Polychlorinated Biphenyl Ethers (PCBs)	PCBs are lipophilic toxicants into adipocytes. In particular, the degree of halogenation or the number and position of chlorine substituents on PCBs affects their uptake and accumulation in adipocytes [119].	Predisposition to obesity, metabolic disorders (disruption of adipose tissue function)
		Organophosphate pesticides	Chlorpyrifos	Chlorpyrifos can cause an increasing number of differentiate 3T3-L1 adipocytes and the capacity for storage of lipid droplets due to up-regulation of transcription factors CCAAT/enhancer binding protein α (C/EBP α) and PPARγ, which is accompanied by significantly higher expression of fatty acid-binding protein 4 (FABP4) adipokin [86].	Metabolic disorders, obesity
			Diazinon	Via inhibition of acetylcholinesterase, diazinon elicits neurotoxicity, significantly induces protein expression of transcription factors CCAAT-enhancer-binding proteins α (C/EBP α) and PPARγ as well as their downstream proteins, fatty-acid synthase (FASN), acetyl CoA carboxylase, lipoprotein lipase, adiponectin, perilipin, and fatty-acid binding protein 4 [89].	Obesity, neurotoxicity
		Organochlorinated pesticides	Dichlordifenyltrichloretan (DDT), Dichlorenthylendichlordiphenyldichlorethylen (DDE)	DDT, DDE can cause disruption of endocrine control, glucose intolerance, dyslipidemia, and hyperinsulinemia [120].	Acute exposure causes harm to the central nervous system, while chronic exposure can result in liver cancer, obesity, harm for the fetus and fertility and increased risk of Type 2 diabetes
		Other environmental pollutants	Benzo[a]pyrene	It can be originator cytotoxicity and expression of inflammation markers [121].	Predisposition to obesity, non-alcoholic fatty acid disease, asthma, hepatic steatosis
			Fine Particulate Matter (PM_2.5_)	PM_2.5_ may cause adipose tissue inflammation [103].	Risk of obesity, predominantly in the male population
			Triclosan	Animal studies show a correlation between high levels of triclosan and estrogens, androgens and thyroid hormones [100].	Risk of obesity
	Additives	Monosodium glutamate		It induces the secretion of glucagon-like peptide-1, a hormone controlling appetite and satiety, and/or antagonization of the androgen receptor, act as a neurotoxic agent on hypothalamic arcuate nucleus and lead to obesity [107,108].	Obesity
		Carrageenan		Is able to affect glucose tolerance, increase insulin resistance and inhibit insulin signaling in in vivo mouse liver cells and human HepG2 cells, promote significant changes in gene expression related to metabolism and lowering of adipokine genes, as well as lipogenesis, absorption, and transport of lipids. Adipolysis and oxidation increase [111,112].	Predisposition to obesity
		Antioxidants		Consuming antioxidant additives might lead to lower leptin secretion and contribute to the obesogenic environment [113,114].	Predisposition to obesity

## Data Availability

Not applicable.

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
