# Peer review of "Obesogens in Foods"

_biomolecules, 2022, doi:10.3390/biom12050680_

Round 1

Reviewer 1 Report

This review focuses on environmental xenobiotics – obesogens – that end up in our food and through a variety of mechanisms contribute to the development of obesity. In general, this is a well-written, although superficial review.

Major concerns:

  1. There are additional obesogens such as artificial sweeteners, trans fats, and alkyl phenols from plasticizers in food that should be included and discussed.
  2. In general, the discussion of how each obesogen contributes to the development of obesity is very superficial, and not comprehensive of the literature.
  3. The statement of “possible health impact” in the title is too neutral/general. The “possible health impact” of obesogens is obesity, as the word “obesogens” would suggest. Changing this part of the title to “mechanisms of pathophysiology” or something along these lines would be more declarative.
  4. Page 3: Please expand the discussion of adipose tissue to include the different types of adipose tissues. Presumably each obesogen has different effects on white and brown adipose tissue function.

Minor comments:

  1. Page 2, Lines 57-59: Please provide a reference for the “current methodological guidelines for systematic reviews” used here. What search terms were used for the systematic review?
  2. Page 2, Line 63: Please describe how inter- and intra-study consistency was addressed.
  3. Page 3, Lines 96-98: Please delete these sentences as they do not pertain to the review material.
  4. Page 5: Please spell out “PCB-126” at first use.
  5. Page 5: Please expand the discussion of how assays of 3T3-L1 cells are used to study the obesogenic potential of xenobiotics.
  6. Page 6: Please delete Line 223.
  7. Page 6: In the section discussing fructose, the first and last paragraphs should be combined for more logical flow.
  8. Page 6, Line 232: In the sentence beginning “Due to different metabolism”, different from what? Please provide more details for this sentence/section.

Reviewer 2 Report

Concerns:

  1. References need to be added after some sentences e.g. after lanes 55, 239.
  2. Conclusions are very short: implications of management of obesogens, as well as the discussion of  further education about exposure and prevention need to be added.
  3. All text needs to be checked for typos and dots which were placed before references.
